# Measurement-induced phase transition in teleportation and wormholes

**Alexey Milekhin[1⋆] and Fedor K. Popov[2]**

**1** University of California Santa Barbara, Physics Department, Santa Barbara, CA, 93106, USA
**2** CCPP, Department of Physics, NYU, New York, NY, 10003, USA

⋆ milekhin@ucsb.edu

## Abstract

We demonstrate that some quantum teleportation protocols exhibit measurement induced phase transitions in Sachdev–Ye–Kitaev model. Namely, Kitaev–Yoshida and Gao–Jafferis–Wall protocols have a phase transition if we apply them at a large projection rate or at a large coupling rate respectively. It is well-known that at small rates they allow teleportation to happen only within a small time-window. We show that at large rates, the system goes into a new steady state, where the teleportation can be performed at any moment. In dual Jackiw–Teitelboim gravity these phase transitions correspond to the formation of an eternal traversable wormhole. In the Kitaev–Yoshida case this novel type of wormhole is supported by continuous projections.


# 1  Introduction

In the past few years, it was observed that hybrid quantum dynamics (i.e. when a quantum system undergoes unitary evolution and is simultaneously subject to measurements or projections) exhibits quite a rich structure [1–16]. In this situation, the von Neumann entropy of the steady state can change between area-law and volume-law, a phenomenon known as Measurement-Induced Phase Transition (MIPT) [1–4]. The presence of a MIPT can be related to an emergent quantum error correction [17–19]. A similar phase transition can also happen between two volume-law phases [20]. Moreover, an MIPT was observed in connected correlation functions [21] or a charge distribution [22]. Recently, measurement dynamics was shown to be useful in proving [23] the NLTS (no low energy trivial states) conjecture [24] and diagnosing the intrinsic sign problem of quantum states [25].

A common feature of all these results is that the phase transition is diagnosed by a quantity that is non-linear in the density matrix. This happens because measurements/projections introduce a lot of energy into the system, resulting in the typical emergent steady-state having infinite temperature, which makes all linear observables trivial. We propose a setup where the system does not heat to an infinite temperature, and the phase transition is then diagnosed by an anti-commutator which is linear in the density matrix.[1] We refer to [6,7] for other ways to avoid infinite heating.

The MIPT has been studied in the quantum field theory (QFT) and gravity context in some recent papers [27–29]. Nonetheless, in the QFT setup MIPTs are mostly overlooked. The main reason is that the heating problem becomes even more severe: in continuum QFT naive application of local measurements lead to ultraviolet (UV) divergences (as is always the case when, in continuum QFT, we operate with quantities localized within "sharp regions"). To resolve this issue, we will develop path-integral techniques to study "weak projections", which consist of coupling the system to an auxiliary qubit, letting them to interact for a certain time and then measuring the auxiliary qubit. In particular, we keep only one of the measurement outcomes. This setup is known as measurement with post-selection, forced measurement or just projection, we will use these terms interchangeably. We would like to stress that it is a physical operation which can be performed in a laboratory setting, albeit at an expense of exponentially many samples: if the probability of obtaining a single desired measurement outcome is $p$, then simulating the system for time $t$ will roughly require $1/p^t$ samples. We will see that in our case this is mathematically equivalent to performing an Euclidean time-evolution. Euclidean evolutions are well-defined even in a continuum QFT.

Although in this paper we study Sachdev–Ye–Kitaev (SYK) model [30–33], which is a quantum-mechanical model for which genuine (non-weak) projections are well-defined, we still restrict ourselves to weak projections. This will allow us to work within the low-energy sector, where a lot of analytical tools are available. Moreover, this low-energy sector has a holographic gravity dual.

We will apply our results to study teleportation protocols. Teleportation protocols address the following problem: suppose we have two subsystems $L$ and $R$ which are initially entangled but otherwise do not interact. How can we efficiently transfer information from one subsystem to another? The most simple quantity which can diagnose teleportation is the anti-commutator[2] between the $L$ and $R$ fermionic operators at times[3]

$$u_1, u_2 : \operatorname{Im} G_{LR}(u_1, u_2) = -i \operatorname{Tr}(\rho_{LR}\{\psi_L(u_1), \psi_R(u_2)\}) \le 1.$$

Teleportation fidelity is proportional to this anti-commutator [34] and in this paper we will concentrate on $\operatorname{Im} G_{LR}$.

---

[1]We refer to [26] which studies MIPT in out-of-time-ordered correlation functions.

[2]Or commutator for bosonic operators.

[3]Following SYK model literature tradition, we denote time by $u$.

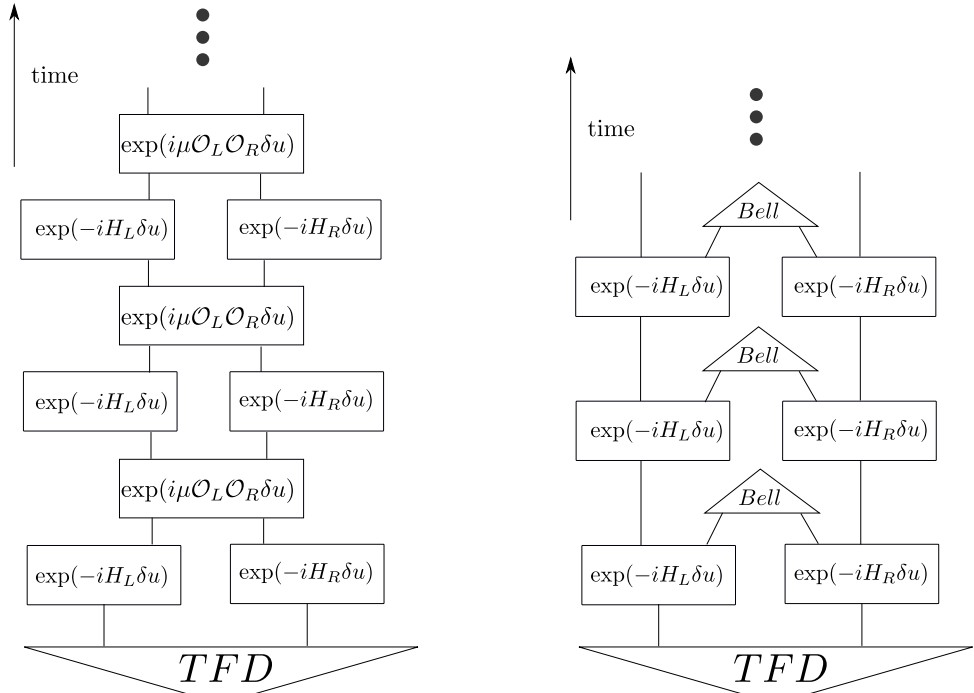

Figure 1: Illustration of our setup. We have two subsystems $L$ and $R$ prepared in the TFD state (3). The goal is to make anti-commutator $\langle\{\psi_L, \psi_R\}\rangle$ non-zero. Left: to this end GJW applies a unitary $e^{i\mu\delta u\mathcal{O}_L\mathcal{O}_R}$, with $\mathcal{O}_{L/R}$ being some hermitian operators. This essentially adds an extra term to the Hamiltonian: $H' = H_L + H_R + \mu\theta(u)\mathcal{O}_L\mathcal{O}_R$. Right: KY takes a subsystem on each side and projects them on the maximally entangled (Bell) state.

Two such protocols are the Kitaev–Yoshida (KY) [35] and the Gao–Jafferis–Wall (GJW) [36–39], which are depicted in Figure 1. We consider left (L) and right (R) subsystems prepared in the thermofield-double (TFD) state, that entangles these subsystems. The total Hamiltonian does not involve L-R interactions, $H = H_L + H_R$, so each subsystem evolves independently. However, TFD state is not stationary under such evolution. Nonetheless, the anti-commutator $\langle\{\psi_L(u_1), \psi_R(u_2)\}\rangle$ is identically zero for all $u_1, u_2$. The goal is to make anti-commutator $\langle\{\psi_L(u_1), \psi_R(u_2)\}\rangle$ non-zero at least for some $u_1, u_2$, allowing the information transfer between $L$ and $R$.

The KY protocol uses a projection for this purpose. In contrast, the GJW protocol applies an extra unitary operator which couples $L$ and $R$ to make $\text{Im}\,G_{LR} = -i\text{Tr}(\rho_{LR}\{\psi_L(u_1), \psi_R(u_2)\})$ large for a finite amount of time. But eventually it decays to zero, prohibiting any information transfer. In the GJW case applying multiple unitary operators at a small rate leads to the same result [36].

*The main question of this paper: is there a qualitative change if these protocols are applied continuously at a high rate? Specifically we study this question in the low-energy (Schwarzian) sector of the SYK model. We indeed find a phase transition to a new steady state when we apply unitary operators or projections at a high rate. In this new phase, teleportation fidelity* $\text{Im}\,G_{LR}$ *stays finite indefinitely.* In the dual gravity picture it means creating an eternal traversable wormhole. In the KY case we will use weak projections in order to avoid heating the system up. The GJW protocol is governed by a Hamiltonian so this problem does not arise.

The whole paper can be summarized by Figures 2 and 3. We turn on the protocols at times $u = 0$, insert a message at $u = u_1 > 0$ and probe with $\text{Im}\,G_{LR}(u_1, T)$ if it reached the other side at $u = T$. In the GJW case we denote two-sided coupling by $\mu$ and for the KY protocol we

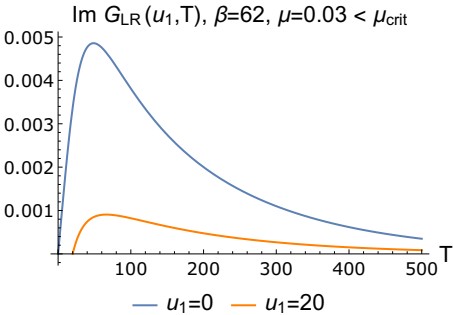

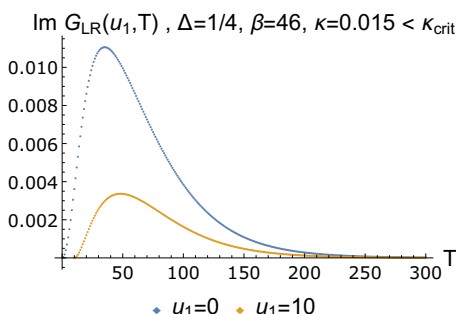

Figure 2: Transfer rate in a slightly perturbed TFD. The coupling/projections are turned on at $u = 0$. A message is inserted at $u = u_1$ and we probe its presence on the other side at $u = T$, hence we plot $\text{Im}\, G_{LR}(u_1, T)$. Left: unitary GJW dynamics. Right: KY protocol with projections (forced measurements). The results are very similar: there is a small transfer amplitude which decays with time. Injecting a message at later times (bigger $u_1$) results in even weaker transition amplitude.

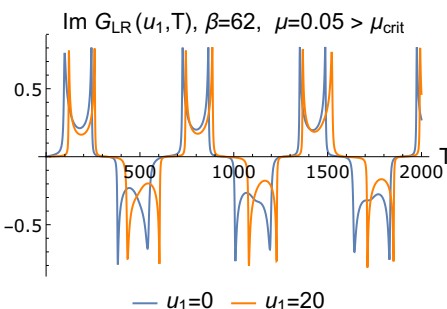

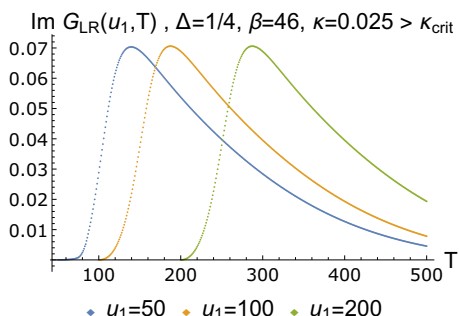

Figure 3: Transfer rate in the strong coupling phase. In both cases we see a dramatic increase in the transfer rate. Left: unitary GJW protocol. Once a message is inserted it bounces back and forth leading to revivals. It does not matter when (different $u_1$) it is inserted. Right: KY protocol with projections (forced measurements). Again it does not matter when a message is inserted. However, there are no revivals.

denote the projection rate[4] $\kappa$. Figure 2 illustrates the phase with small $\mu$ or $\kappa$. The results are very similar for the GJW and KY: initially $\text{Im}\, G_{LR}$ is non-zero, but then exponentially drops to zero if we try to insert a message at later times $u_1$. This is essentially the GJW computation. Again, note that the interaction term $\mu \mathcal{O}_L \mathcal{O}_R$ is constantly on for all times $u > 0$. We can greatly enhance the rate if we insert a message at some specific negative time $u_1$ before the protocols are turned on at $u = 0$. However, in this paper we are interested if we can teleport at later times.

At large $\mu$ or $\kappa$ a qualitative change occurs - Figure 3. We can insert a message at any time $u_1 > 0$ and it will reach the other side with fidelity that does not depend on the time $u_1$. This is the only common feature between the GJW and KY in the strong coupling regime. In the GJW case the message bonces back-and-forth without dissipation, leading to revivals. Maximal value of $\text{Im}\, G_{LR}$ is of order 1, and it has a weak dependence on $\mu$ and initial temperature $\beta$. In the KY case a message reaches the other subsystem and then dissipates: $\text{Im}\, G_{LR}$ decays to zero at late times. Its maximal value does not depend on initial temperature, but it depends on $\kappa$.

The strong coupling phase in the GJW case was discussed by Maldacena and Qi [40]. The

---

[4]That is, a projection is applied every $1/\kappa$ time interval.

SYK model is dual [32, 41] to the two-dimensional Jackiw–Teiltelboim gravity where this phase represents an eternal traversable wormhole. Our contribution is that it is possible to reach this phase starting from TFD state, which represents two entangled black holes. This can be done easily (without coupling to external systems), but it requires a finite $\mu$ coupling:[5] the transition happens only if $\mu$ is large enough. We will obtain the corresponding critical value analytically. Such behavior was conjectured for higher-dimensional black holes [43]. A similar problem was addressed by Lensky and Qi [44] in large $q$ SYK, although the phase transition is absent there, we discuss this in Section 3.3. The strong coupling phase of the KY protocol represents a novel type of a wormhole supported by projections. Wormhole length can be diagnosed by correlator $-\log G_{LR}(T, T)$ at coincident time points. For TFD state it grows linearly with time, reflecting the growth of the Einstein–Rosen bridge. We find that this quantity becomes constant for the KY wormhole, similarly to Maldacena–Qi (MQ) wormhole. Finally, let us point out that in the GJW/MQ case a similar transition can happen [45] even if the $L, R$ subsystems initially unentangled.[6] However, it requires coupling to an external bath and the analysis has to be performed in full SYK, not just in the low-energy Schwarzian sector.

Finally, let us address one apparent difference between the GJW/MQ traversable wormhole and the KY one. The most prominent feature of the GJW/MQ wormhole is the presence of strong revivals in the correlation function. From the Figure 3 it is not obvious if they are present in the KY case.[7] We claim that they are present. In Section 4.3 we discuss the analytical approximation to the KY wormhole. We find that at late times the correlation function can be approximated by

$$G_{LR}(u_1, u_2) = \text{const} \frac{1}{\cos^{2\Delta}\left(\frac{e^{\phi_*}(u_1 - u_2)}{2}\right)}, \tag{1}$$

where $e^{\phi_*}$ is complex. Hence the correlation functions oscillates, but the oscillations are damped due to decoherence caused by the projections. Figure 9 shows the behavior at very late times demonstrating revivals.

The paper is organized as follows. In Section 2 we present a very general discussion of weak projections. In Section 3 we discuss the GJW protocol in SYK and setup the necessary machinery. In Section 4 we combine the two together to discuss the non-unitary KY dynamics. We will discuss some of the open questions in the Conclusion.

## 2 Projections at low energies

Let us consider the following setup. Suppose that we have a Hamiltonian $H$ acting on $N/2$ qubits, with energies $E_n$ and eigenvectors $|n\rangle$. We double the system, such that we have two identical subsystems, which we denote as $L$ and $R$. They have $N$ qubits in total. We can use Jordan–Wigner transformation to map them to $2N$ Majorana fermions $\psi_L^i, \psi_R^i$, $i = 1, \ldots, N$. These operators square to one and anticommute

$$\left\{\psi_\alpha^i, \psi_\beta^j\right\} = \delta^{ij}\delta_{\alpha\beta}, \quad \alpha, \beta = L, R, \quad i = 1, \ldots, N. \tag{2}$$

---

[5]Equations are simple enough to have an exact solution for any $\mu$. We believe the transition to an eternal wormhole cannot be seen in the perturbation theory in $\mu$ because above the critical $\mu$ the classical solution completely changes its behavior. The equations can be mapped to a classical particle moving in a potential. If $\mu$ is large enough the particle trajectory becomes trapped rather than running away to infinity. Such qualitative change of behavior cannot be treated as a small perturbation. We refer to [42] for the perturbative discussion of the bulk dilaton.

[6]This models the scenario when two unentangled black holes evaporate and exchange radiation.

[7]We thank the anonymous reviewers for raising this question.

The system is prepared in the thermofield-double (TFD) state:

$$|TFD\rangle = \sum_{E_n} e^{-\beta E_n/2} |n_L\rangle \overline{|n_R\rangle}, \tag{3}$$

where bar denotes time-reversal operation. The evolution is governed by a Hamiltonian $H = H_L + H_R$, that does not include interaction between these two subsystems. In order to teleport information between $L$ and $R$ we add a certain interaction between the sides.

First we address the question of how to add a projection operation. Ideally, we are interested in adding a projection on a maximally mixed state, which can be written as follows

$$\Pi = \prod_{j=1}^{N} \Pi_j = \prod_{j=1}^{N} \frac{1}{2} \left( 1 - i \psi_L^j \psi_R^j \right). \tag{4}$$

The projection $\Pi$ makes the two subsystems maximally entangled. This would lead immediately to an infinite temperature. Instead, we want to act very softly and make the process continuous. We will use Schwinger-Keldysh techniques in order to formulate this in the path integral language.

Let us introduce a probability $\kappa$ of performing a projection (measurement with postselection) per unit time. Hence the precise evolution equation is

$$\widetilde{\rho}(u+du) = i[H, \widetilde{\rho}(u)]du + \widetilde{\rho}(u)(1 - 4N\kappa du) + \kappa du \sum_j (1 - i\psi_L^j \psi_R^j)\widetilde{\rho}(u)(1 - i\psi_L^j \psi_R^j). \tag{5}$$

Here we are working with unnormalized density matrix $\widetilde{\rho}$. Observables in this case are evaluated as

$$\langle \mathcal{O} \rangle = \frac{\text{Tr}(\widetilde{\rho}\mathcal{O})}{\text{Tr}\widetilde{\rho}}. \tag{6}$$

Physically, these equations correspond to the following physical situation. First, we have multiple copies of the system. Each copy is evolved separately. Each time interval $du$ the system does not change with probability $1 - 4N\kappa du$. Otherwise, with probability $4N\kappa du$ we pick a random $j$ and measure $i\psi_L^j \psi_R^j$. If the outcome is $-1$, we disregard this copy altogether. If the outcome is $+1$ then we proceed with the unitary evolution. Measuring an observable involves taking the number of copies to infinity and averaging over all copies where the outcome is $+1$.

One has to normalize the observables by diving by $\text{Tr}\widetilde{\rho}$. In a generic disordered system it might be hard to do the disordered averaging of such ratios. However, in SYK it is known that for large $N$ the interaction between replicas is suppressed by $1/N$ and we can do the averaging in the following way [31,32]:

$$\left\langle \frac{\text{Tr}(\widetilde{\rho}\mathcal{O})}{\text{Tr}\widetilde{\rho}} \right\rangle \approx \frac{\langle \text{Tr}(\widetilde{\rho}\mathcal{O}) \rangle}{\langle \text{Tr}(\widetilde{\rho}) \rangle} + \mathcal{O}\left(\frac{1}{N}\right). \tag{7}$$

Hence we can separately write a path integral for numerator and denominator.

This evolution does not preserve trace, but still all correlators satisfy usual hermicity conditions. The advantage of dealing with $\widetilde{\rho}$ is that it has a nice Schwinger-Keldysh path-integral representation. It can be obtained as follows. First, the Hamiltonian evolution acts on density matrix $\rho$ in the following way:

$$\rho \to e^{-iHu} \rho e^{+iHu}. \tag{8}$$

And in infinitesimal form:

$$\rho \to \rho - iHdu\rho + i\rho Hdu. \tag{9}$$

Each $e^{iHu}$ can be formulated as a path integral. This is the origin of $\pm$ in Schwinger–Keldysh (SK) formalism. $e^{-iHu}$ corresponds to (forward) plus part, and $e^{+iHu}$ to (backward) minus part. They also have different signs:

$$Z_{SK} = \int D\psi_+ D\psi_- \; e^{iS_+ - iS_-} . \tag{10}$$

Without extra insertions in the path integral we identify $+$ fields with $-$ fields and get $Z_{SK} = 1$, hence the trace is conserved. Now comparing our master eq. (5) with the standard Hamiltonian dynamics, eq. (9), we see that $\pm$ parts, coming from the forward and the backward time evolutions of the density matrix, do not cancel, and the trace is not conserved.

The resulting action coming from eq. (5) is[8]

$$Z_{SK} = \int D\psi_\pm \; \exp\left( iS_+ - iS_- - \kappa \sum_j \int du \; \left( i\psi^+_{R,j}\psi^+_{L,j} + i\psi^-_{L,j}\psi^-_{R,j} + \psi^+_{R,j}\psi^+_{L,j}\psi^-_{L,j}\psi^-_{R,j} \right) \right) . \tag{11}$$

This evolution projects fermions at a steady rate $\kappa$, instead of projecting them all at once, but it still introduces a lot of energy into the system. This can be traced to the quartic fermionic term in the above action. Moreover, although such term is perfectly permissible for quantum mechanical systems, such as SYK, it does not make sense in a continuum QFT, as we have to take product of operators at coincident points, $\psi^+_{R,j}\psi^-_{R,j}$.

In order to circumvent this problem we consider weak projections. It means that we couple $\psi^j_L, \psi^j_R$ to an auxiliary qubit $|0_A\rangle$, they undergo evolution together and after that we measure and post-select the state of this auxiliary qubit. Mathematically, we start from the density matrix

$$|0_A\rangle\langle 0_A| \otimes \rho , \tag{12}$$

it undergoes the following unitary evolution

$$|0_A\rangle\langle 0_A| \otimes U^\dagger_{00}\rho U_{00} + |0_A\rangle\langle 1_A| \otimes U^\dagger_{01}\rho U_{01} + |1_A\rangle\langle 0_A| \otimes U^\dagger_{10}\rho U_{10} + |1_A\rangle\langle 1_A| \otimes U^\dagger_{11}\rho U_{11} . \tag{13}$$

Then we post select on

$$|0_A\rangle\langle 0_A| \otimes U^\dagger_{00}\rho U_{00} . \tag{14}$$

Operators $U_{ab}$, $a, b = 0, 1$ act on $\psi^j_L, \psi^j_R$ Hilbert space (which is 2 dimensional). They do not have to be unitary. Specifically, we want $U_{00}$ to be

$$U_{00} = e^{-\kappa du S_j}, \; S_j = 1 + i\psi^j_L\psi^j_R , \tag{15}$$

such that it damps $i\psi^j_L\psi^j_R = 1$ subspace. However, the complete $U$ operator,

$$U = \left( \begin{array}{c|c} U_{00} & U_{01} \\ \hline U_{10} & U_{10} \end{array} \right) , \tag{16}$$

has to be unitary. We can easily find the missing $U_{01}, U_{11}$ from unitarity:

$$U = \left( \begin{array}{cc|cc} 1 & 0 & 0 & 0 \\ 0 & e^{-2du\kappa} & 0 & \sqrt{1 - e^{-4du\kappa}} \\ \hline 0 & 0 & 1 & 0 \\ 0 & \sqrt{1 - e^{-4du\kappa}} & 0 & -e^{-2du\kappa} \end{array} \right) . \tag{17}$$

---

[8]Possible extra constant $c$ in $\widetilde\rho \to c du \widetilde\rho$ does not matter, as it will be eventually cancelled by trace normalisation.

The above procedure can be performed for any pair index $j$. The upshot is that the system undergoes the following Euclidean evolution:

$$\widetilde{\rho}(u + du) = e^{-\kappa S du}\widetilde{\rho}(u)e^{-\kappa S du}, \qquad S = i\sum_j \psi_L^j \psi_R^j. \tag{18}$$

Such evolution is obviously hermitian and completely positive. The corresponding action is

$$iS_{\text{weak proj}} = -\kappa \sum_j \int du \left(i\psi_{R,j}^+ \psi_{L,j}^+ + i\psi_{L,j}^- \psi_{R,j}^-\right). \tag{19}$$

We will denote $e^{-\kappa S}$ as $\Pi_\kappa$. This term was recently studied in [12] for Brownian SYK. However, in that paper it has a completely different origin. For comparison, MQ interaction looks like

$$iS_{MQ} \propto \mu \sum_j \int du \left(\psi_{L,j}^+ \psi_{R,j}^+ - \psi_{L,j}^- \psi_{R,j}^-\right). \tag{20}$$

Notice an extra $i$ and a difference in the relative sign between $\pm$ parts. That leads to unitary evolution that preserves energy and trace.

## 3 Turning TFD into an eternal traversable wormhole with Gao–Jafferis–Wall

### 3.1 The setup

In this Section, we study the SYK model out of equilibrium when we apply an extra unitary $\exp(i\mu\mathcal{O}_L\mathcal{O}_R\delta u)$ on top of the standard Hamiltonian dynamics - Figure 1. $\mathcal{O}_{L/R}$ are any operators on the left/right. We will review the formalism of Maldacena and Qi [40] which we will later apply in Section 4 when we discuss dynamics under continuous projections. Most of the equations in this Section, except eqns. (25), (26), are valid for arbitrary time-dependent coupling $\mu(t)$. Our contribution is solving them after $\mu$ is suddenly turned on and understanding the fate of the system.

The total Hamiltonian is

$$H = H_{SYK,L} + H_{SYK,R} + i\mu\theta(u - u_0)\mathcal{O}_L\mathcal{O}_R, \tag{21}$$

$$H_{SYK,L/R} = i^q \sum_{i_1...i_q} J_{i_1...i_q}\psi_{L/R}^{i_1}...\psi_{L/R}^{i_q}, \quad \langle J_{i_1...i_q}^2\rangle = J^2\frac{(q-1)!}{N^{q-1}}, \tag{22}$$

note that the disorder couplings $J_{ijkl}$ are the same for both subsystems. Initially the system is prepared in the TFD state. We wrote the microscopic Hamiltonian explicitly, but we would not really need it, as we will be concentrated on the low energy sector, that is governed by the Schwarzian action. The only important thing is that both $L$ and $R$ subsystems are subject to the same disorder. The case of instantaneous insertion of $\mu\mathcal{O}_L\mathcal{O}_R$ was extensively studied in [46, 47]. Here we address the question of what happens if we switch on a large $\mu$ at $u = u_0$ and keep it on.

One of the most interesting features of SYK is that in the limit of large $N$ and low energies it develops an approximate conformal and reparametrization symmetry. The corresponding action for reparametrizations is given by the Schwarzian action. Equations of motion of the Schwarzian are local and corresponding classical solutions determine all correlation functions in the leading order in $1/N$. This is true for any state, even the ones that are out of equilibrium

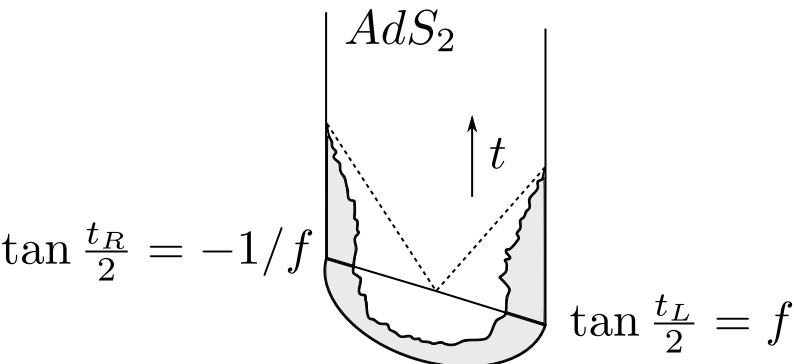

Figure 4: Analytical continuation from TFD state.

[48]. This is a great simplification, as in general mean-field equations are highly non-local. As a side note, there are SYK models where low-energy effective action is dominated by a non-local action for reparametrizations [41, 49, 50], but here we concentrate on the Schwarzian case.

In the low energy sector of SYK all correlation functions depend only on reparametrizations $f(u)$, that determine two-point functions of operators of dimension $\Delta$ as

$$\langle W_\Delta(u_1)W_\Delta(u_2)\rangle = \left(\frac{f'(u_1)f'(u_2)}{(f(u_1)-f(u_2))^2}\right)^\Delta . \tag{23}$$

For example, elementary fermions have dimension $\Delta = 1/q$. The reparametrizations $f(u)$ are governed by the following action

$$S/N = -\frac{\alpha_s}{J}\int d\tau \ \mathrm{Sch}(f(\tau),\tau), \quad \mathrm{Sch}(f(\tau),\tau) = \left(\frac{f''}{f'}\right)' - \frac{1}{2}\left(\frac{f''}{f'}\right)^2 . \tag{24}$$

We put the (dimensionful) Schwarzian coupling to be $\alpha_s/J = 1$, so we will measure everything in terms of this coupling. In the TFD state, $L$ or $R$ separately look thermal. The corresponding Euclidean solution is $f_{L,R}(\tau) = \tan\left(\frac{\pi\tau}{\beta}\right)$. One can easily check that it is a solution of the Schwarzian theory.

In this paper we will be dealing mostly with Lorentzian solutions parametrized by time $u$. Later when we return to solutions on Euclidean circle we will simply analytically continue to $-iu$. Since we have Lorentzian evolution of two independent SYK dots we need to use two reparametrizations to describe each of the systems. If we depict $\exp(-H\beta/2)$ in the definition of TFD as Euclidean evolution along a semicircle then the two trajectories in Lorentzian are obtained from two endpoints of the semicircle - Figure 4. The corresponding solutions are:[9]

$$t_L = t_R = 2\arctan\tanh\frac{\pi u}{\beta} , \tag{25}$$

and two-sided Green function reads as

$$G_{LR} = \left(\frac{\pi/\beta}{\cosh\left(\frac{\pi(u_1+u_2)}{\beta}\right)}\right)^{2\Delta} . \tag{26}$$

Obviously, $\mathrm{Im}\,G_{LR} = 0$ which reflects the fact that we cannot transfer any information between the sides. Geometrically it is linked to the fact that $t_L, t_R$ have finite range, namely

---

[9]In geometrical terms, $u$ is (physical) boundary time, $f$ is $AdS_2$ Poincare coordinate time and $t_{L,R}$ are global $AdS_2$ times.

$t_{L,R} \in [-\pi/2, \pi/2]$. It means that a signal cannot propagate from one side to the other - Figure 5 (a).

If we add an interaction $\mu \neq 0$, the picture changes. Again at low temperatures we can use reparametrizations $f_{L,R}$ to describe the behaviour of the system. If $\mu$ is small enough we can take the leading perturbative answer $\mu\langle\mathcal{O}_L\mathcal{O}_R\rangle$ and reparametrize it:

$$S/N = -\int du \ \text{Sch}\left(\tan\frac{t_L}{2}, u\right) - \int du \ \text{Sch}\left(\tan\frac{t_R}{2}, u\right) + \mu\theta(u-u_0)\int \ du \left(\frac{t'_L t'_R}{\cos^2\frac{(t_L-t_R)}{2}}\right)^\Delta. \quad (27)$$

The last term is obtained from eq. (23) by taking $f(u_1) = \tan(t_L/2)$ and $f(u_2) = -1/\tan(t_R/2)$. In this choice of coordinates, correlation function are determined by [10]

$$G_{LL}(u_1, u_2) = i^\Delta \left(\frac{t'_L(u_1)t'_L(u_2)}{\sin^2(t_L(u_1)/2 - t_L(u_2)/2)}\right)^\Delta ,$$

$$G_{LR}(u_1, u_2) = \left(\frac{t'_L(u_1)t'_R(u_2)}{\cos^2(t_L(u_1)/2 - t_R(u_2)/2)}\right)^\Delta . \quad (28)$$

However, in terms of dynamics this is not the full story. The action (23) is redundant: any $SL(2,R)$ linear-fractional transformation on $f$ will leave physical Green function invariant. Hence in the Schwarzian theory the global $SL(2,R)$ must be gauged [41]. The total $L+R$ charges $Q_{0,\pm1}$ must be set to zero. J.Maldacena and X.Qi showed that $Q_{\pm1}$ constraints can be satisfied by the symmetric solution $t_L(u) = t_R(u) = t(u)$. The remaining $Q_0$ constraint can be recast as

$$0 = Q_0 = 2e^{-\phi}\left(-\phi'' - e^{2\phi} + \mu\Delta e^{2\Delta\phi}\right), \quad \phi(u) = \log t'(u), \quad (29)$$

that provides the equation of motions for the $t'(u)$ and could be considered as a motion of a particle in one dimensional potential. Quantity $\ell = -\phi$ is proportional to $-\log G_{LR}(u,u)$. It characterises the distance between the trajectories in $AdS_2$ - Figure 5.

## 3.2 Turning on the interaction

In the previous subsection we introduced the MQ equations for studying two interacting SYK models. This subsection contains new results about the time-dynamics of two interacting SYK models. At low energies the dynamics of the system boiled down to one simple equation (29), that could be understood as a simple one-dimensional classical mechanics problem. This problem possesses some integrals of motions. For instance, we can find that "energy" is conserved

$$E = \phi'^2 + e^{2\phi} - \mu e^{2\Delta\phi}, \quad (30)$$

and the corresponding dynamics depends on the sign of $E$.

- Positive or zero[11] energy $E \geq 0$. In this case we have a run-away solution at infinity, $\phi = -\gamma u$. It implies that $t(u) \sim e^{-\gamma u}$. Hence the range is finite. This is illustrated by Figure 5 (b). TFD becomes traversable for a finite amount of time.

- For negative energy $E < 0$ there is a qualitative change. The particle is confined in a one-dimensional potential and oscillates near the minimum of the potential. Obviously, we can state that $\phi_{\min} < \phi < \phi_{\max}$. Since $\phi = \log t'$ we conclude that $t'$ is always positive and $t$ grows all the way to infinity. We have an eternal traversable wormhole,

---

[10] We are omitting extra $\Delta$-dependent normalization. For elementary fermions in SYK one has to multiply these expressions by $c_\Delta = J^{-2\Delta}((1/2 - \Delta)\tan\pi\Delta)^\Delta$.

[11] The below analysis covers only positive energy. A slightly more elaborate analysis shows that for zero energy $t' \propto 1/u^{1/\Delta}$. Hence for $\Delta < 1$ the range is finite too.

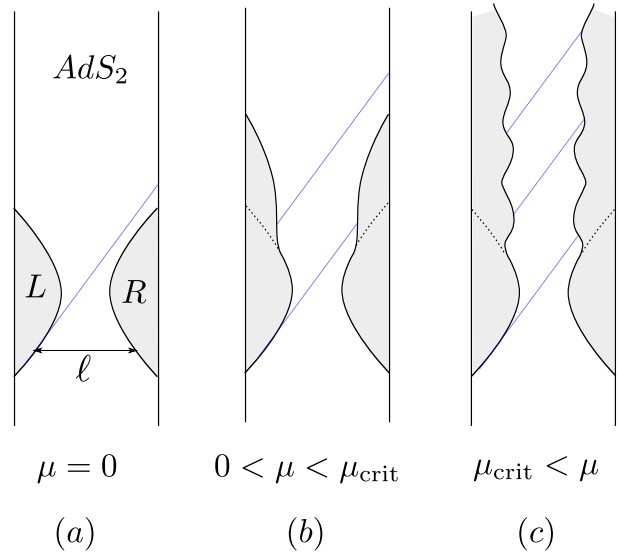

Figure 5: (a) No interaction between the sides, we have a TFD solution. (b) Non-zero $\mu$ is switched on, but it is not big enough to make the wormhole eternally traversable. (c) For large enough $\mu$ we get an eternal traversable wormhole.

Figure 5 (c). To illustrate the behavior of $G_{LR}$ we can fine-tune the interaction parameter $\mu$ to make $t'$ constant. Then the solution is simply [40]:

$$G_{LR}(u_1, u_2) \propto \left( \frac{1}{\cos(t'(u_1 - u_2))} \right)^{2\Delta},$$

$$\text{Im}\, G_{LR}(u_1, u_2) \propto \theta\left( \cos(t'(u_1 - u_2)) \right) \sin \pi \Delta \left( \frac{1}{\cos(t'(u_1 - u_2))} \right)^{2\Delta}, \qquad (31)$$

this propagator has a branch cut that opens at $u_1 - u_2 = \frac{\pi}{2t'}$ and therefore has non-zero imaginary part. The fact that it blows up indicates that the information transfer is too strong, effectively leading to operators colliding and therefore inapplicability of the low energy approximation. The actual finite value (of order 1) of $G_{LR}$ can be obtained in large-$q$ SYK (next Section) or by numerically solving the corresponding systems of Dyson-Schwinger equations, but that is outside of the scope of this paper.

These different cases take place depending on $\mu$ and time $u_0$ when the interaction was turned on. By usual arguments we conclude that the functions $\phi, \phi'$ must be continuous. It allows to compute the effective energy as

$$E = \phi'^2_{TFD}(u_0) + e^{2\phi_{TFD}(u_0)} - \mu e^{2\Delta \phi_{TFD}(u_0)}. \qquad (32)$$

Using eq. (25) we find

$$E = \frac{4\pi^2}{\beta^2} - \mu \left( \frac{2\pi/\beta}{\cosh\left[ \frac{2\pi u_0}{\beta} \right]} \right)^{2\Delta}. \qquad (33)$$

As we explained above the phase transition happens when $E < 0$. At early times, when the correlation between the sides is still big, we only need $\mu^* \sim \beta^{2\Delta - 2}$. As was announced in the Introduction, for a fixed initial state it requires a finite $\mu$ coupling, determined by $E = 0$, to create an eternal traversable wormhole. For example, static MQ wormhole corresponds to the absolute minimum of the energy (30), which is $E_{min} = 1/(\Delta \mu)^{1/(1-2\Delta)} - \mu/(\Delta \mu)^{\Delta/(1-2\Delta)}$. It

is possible to reach it only if the interaction is switched on at $u_0 = 0$. In general for a fixed $\mu$, (33) is always greater than $E_{min}$.

It is very important to check that we are inside the perturbative regime. The actual perturbative parameter is $\mu\beta^{1-2\Delta} \ll 1$ [45], whereas we only require $\mu^* \sim \beta^{2\Delta-2}$. So low-energy approximation is applicable for $\beta \gg 1$ (in dimensionless units) as expected.

We can also repeat this analysis if the initial state is an excited TFD. It means that $SL(2,R)$ charges are not zero.[12] For simplicity we assume $L \leftrightarrow R$ symmetry. In this case the only information about the excitations is encoded in $Q_0 > 0$. Now the motion is controlled by the effective Hamiltonian or energy:

$$E = \phi'^2 + V(\phi), \quad V(\phi) = Q_0 e^\phi + e^{2\phi} - \mu e^{2\Delta\phi} . \tag{34}$$

For $\Delta < 1/2$ the story is essentially the same: potential has only one global minima and the energy is negative there. Hence having a bounded trajectory (traversable wormhole) is equivalent to having negative energy.

However, for $\Delta > 1/2$ there is a new phenomenon: energy can be positive, but the classical motion is confined within a finite $\phi$ region around a local minima, yielding an eternal traversable wormhole. It happens because now the potential has a local maxima. But still we have $V(\phi) \to 0, \phi \to -\infty$. It means that such solution is metastable at finite $N$ and due to the quantum corrections this wormhole will close at some point. And the rate of the decay is non-perturbatively small $N$: $\Gamma \sim e^{-N}$.

### 3.3 Comments on large-$q$ SYK

We should also make a comment about large-$q$ SYK. In this case it is possible to find the correlators at any time separation. However, in this limit the dimension of an elementary fermion operator goes as $1/q$. Hence one ends up with a *long-range* attraction potential $V(\phi) \propto \mu\phi$ in the effective Hamiltonian. In this case there is no phase transition: any small $\mu > 0$ will lead to a wormhole. This setup was studied in [44] and we refer to this paper for technical details. We will borrow their results to study the transition and correlation functions at any energy scale. In order to see a phase transition, the coupling operator must have the form of $\left(\psi_L^j \psi_R^j\right)^k$, where $k/q$ stays finite for $q \to \infty$. The derivation of [44] can be easily applied in this case as well, leading to the following large $q$ analogue of (30):

$$E = -2\cos(p)\sqrt{1 - e^{2\phi}} - \mu e^{2\Delta\phi} , \tag{35}$$

with $\phi, p$ being conjugate variables. Now the correlation functions are determined by a *complex* reparametrization $\chi(u)$ as (compare with (28))

$$G_{LL} \propto i^\Delta \left( \frac{\chi'(u_1)\chi'(u_2)^*}{\sin^2 \frac{\chi(u_1)-\chi(u_2)^*}{2}} \right)^{2\Delta} , \qquad G_{LR} \propto \left( \frac{\chi'(u_1)\chi'(u_2)^*}{\cos^2 \frac{\chi(u_1)-\chi(u_2)^*}{2}} \right)^{2\Delta} . \tag{36}$$

$\chi(u)$ can be easily found from

$$\chi' = \frac{e^{ip}}{\sqrt{e^{2\phi} - 1}} . \tag{37}$$

In the limit of small $p$ and large $\phi$ we recover eq. (30). The analysis of trajectories can be done in the same way as in the previous Section. However, this formalism allows us to compute $G_{LR}$ at any time separation. This is how we obtained the left panels of Figure 2 and Figure 3.

---

[12]We should take into account that to create an excited TFD we should have introduced some operators that create such state. Now if we add contributions to the charges from these operators we get zero.

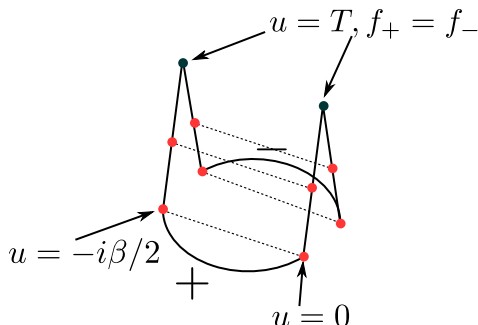

Figure 6: SK contour for projections dynamics. Red dots indicate projections. They couple left and right subsystems, which is indicated by a dashed line. Semicircles prepare TFD state and the vertical parts represent Lorentz-time evolution (they should be drawn as vertical lines instead of wedges and the red dots at the bottom should coincide, but we separated them to make the drawing more clear).

## 4  Projection dynamics

In this Section we will combine the results from the previous two sections and study the dynamics with continuous projections. First we study Kitaev–Yoshida protocol with a single projection. After that we study the continuous dynamics that involve multiple projections.

Before proceeding to the actual computation we should understand how to properly diagnose traversability, that is, how to check that information is transmitted from one subsystem to the other. The most simple thing to do is to insert $\psi_L(u_1)$ at some time $u_1$, perform projection at $u = 0$ and then measure if the information has reached by other qubit $\psi_R(u_2)$. The transmission of the information would result at non-zero anti-commutator of these two operators. Thus the corresponding matrix element is

$$\mathrm{Im}\, G_{LR}(u_1, u_2) = -\mathrm{Im}\, \frac{i\langle TFD|\psi_L(u_1)\Pi_\kappa(0)\psi_R(u_2)\Pi_\kappa(0)|TFD\rangle}{\langle TFD|\Pi_\kappa^2|TFD\rangle}, \tag{38}$$

where $\Pi_\kappa = e^{-\kappa S}$ is the weak projector in eq. (18). SK path-integral (11) naturally produces Green functions normalised this way. Also this intuition can be made precise if one follows the protocol proposed in [34].

Another way to normalize the anti-commutator is

$$\mathrm{Im}\, G_{LR}^{\mathrm{norm}} = -\mathrm{Im}\, \frac{i\langle TFD|\psi_L(u_1)\Pi_\kappa(0)\psi_R(u_2)\Pi_\kappa(0)|TFD\rangle}{\langle TFD|\psi_L(u_1)\Pi_\kappa(0)^2\psi_L(u_1)|TFD\rangle}. \tag{39}$$

This is natural too because we want to normalize the state *after* we inserted $\psi_L$. Unfortunately, we do not know any specific protocols which would measure $G_{LR}^{\mathrm{norm}}$. However, this quantity suggests an enhanced transfer rate compared to (38) so we will study it too.

### 4.1  Boundary conditions

The matrix element $G_{LR}(u_1, u_2)$ (38) was studied by Streicher and Qi [51] for large $q$ SYK in the context of operator growth. In their setup the operator $S$ played the role of the size operator. We will reproduce this result using the low-energy Schwarzian approximation.

The relevant Schwinger-Keldysh contour for computing the correlators (38),(39) is shown on Figure 6. We will need 4 reparametrizations: two Euclidean ones $f_\pm^\beta$ and two Lorentzian ones $f_\pm$. The only non-standard feature of our calculation is that forward $+$ and backward $-$ evolutions do not cancel each other due to the insertion of weak projector operators. Hence

$f_+ \neq f_-$, unlike the usual Hamiltonian evolution.[13] However they still obey some conditions. First, the two-point function must be real, that requires $f_-(u) = (f_+(u))^*$. Secondly, when we do not conduct any more projections the systems should be evolved by a standard Hamiltonian evolution, so $f_-(u) = f_+(u)$ and by demanding continuity we arrive at $f_+(T) = f_-(T)$, where $T$ is a moment of the last weak projection. Moreover, the Schwarzian equations of motion imply that $f, f'$ must be continuous, when the weak projection is inserted. That provides us with another boundary condition. Another case when such conditions arise is when we evaluate expectation values at time $T$. In this case $\pm$ contours merge and we get the same set of conditions.

Also we need to determine the boundary conditions on the thermal circle. Again, because $\pm$-parts of SK contour do not cancel out, the dynamics is not causal so we should not expect to have a standard thermal solution. We want to find $f_\pm$ such that analytic continuation of a solution to Lorentzian signature would respect a $L \leftrightarrow R$ symmetry. Upon analytic continuation to Lorentzian signature, the $AdS_2$ global times $t_{L,R}(u)$ are determined in terms of Poincare time $f_\pm^\beta(u)$ by the following equations:

$$\tan \frac{t_L^\pm(u)}{2} = f_\pm^\beta(u), \qquad \tan \frac{t_R^\pm(u)}{2} = -\frac{1}{f_\pm^\beta(-i\beta/2-u)}. \tag{40}$$

Upon demanding the $L-R$ symmetry we can find a gauge for $f_+^\beta$ such that $t_L(u) = t_R(u)$. This fixes $f_+^\beta$ to be

$$f_+^\beta(u) = \frac{e^{-\alpha u} - iA_+ e^{i\alpha\beta/4}}{A_+ e^{-\alpha u} + i e^{i\alpha\beta/4}}, \tag{41}$$

where $\alpha, A_+$ are some constants, that we will fix later.[14]

## 4.2 Single projection: Recovering Kitaev–Yoshida teleportation

First, let us discuss a dynamics after one single weak projection. It corresponds to a situation with one pair of red dots at the bottom in the Figure 6. Inserting $i\psi_L\psi_R$ at time $u^*$ into the action generates a certain discontinuity in reparametrizations, that could be determined from the equations of motions. Thus we consider the following action

$$S = -\int d\tau \ \text{Sch}(f_L, u) - \int d\tau \ \text{Sch}(f_R, u) + i\kappa \int du \ \delta(u-u^*) \left( \frac{f_L'(u)f_R'(u)}{(f_L(u)-f_R(u))^2} \right)^\Delta, \tag{43}$$

Under the variation of $f_L$ the Schwarzian we get the following equation

$$\delta \, \text{Sch} = -(\text{Sch})' \frac{\delta f_L}{f_L'} = \left[ -\frac{f_L''''}{f_L'^2} + \frac{4f_L''f_L'''}{f_L'^4} - \frac{3f_L''^3}{f_L'^4} \right] \delta f_L, \tag{44}$$

whereas the most singular part of the $\kappa$ term comes from varying $f_L'$:

$$\delta S_{\text{int}} = -\frac{\delta f_L}{f_L'(u^*)} \delta'(u-u^*)\kappa\Delta \left( \frac{f_L'(u^*)f_R'(u^*)}{(f_L(u^*)-f_R(u^*))^2} \right)^\Delta + \frac{\mathcal{A}}{f_L'}\delta(u-u^*), \quad \mathcal{A} = \text{const}. \tag{45}$$

---

[13]This is just the statement that in the semiclassical limit with Hamiltonian evolution Schwinger-Keldysh action is controlled by $f_+ = f_-$. In our case we are in the semiclassical regime because of large $N$.

[14]Simultaneous shift of time on both sides by $2x$, $u_{L,R} \to u_{L,R} + 2x$ generates a gauge transformation of a constant $A_+$:

$$A_+ \to \frac{A\cos x - \sin x}{\cos x + A\sin x}. \tag{42}$$

So the global time shifts in $AdS_2$ are not fixed yet.

Then we arrive at the following equation

$$\text{Sch} = -\delta'(u-u^*)\kappa\Delta\left(\frac{f_L'(u^*)f_R'(u^*)}{(f_L(u^*)-f_R(u^*))^2}\right)^\Delta + \mathcal{A}\theta(u-u^*),\tag{46}$$

that allows us to determine that the discontinuity in the second derivative that would also provide the boundary conditions. Thus we arrive at

$$f_L''(u^*+\epsilon)-f_L''(u^*-\epsilon)=i\widetilde{\kappa}f_L'(u^*),\quad \widetilde{\kappa}=\kappa\Delta\left(\frac{f_L'(u^*)f_R'(u^*)}{(f_L(u^*)-f_R(u^*))^2}\right)^\Delta,\tag{47}$$

for convenience we defined a new parameter parameter $\widetilde{\kappa}$. This boundary conditions together with equations of motions allows us to completely fix the form of the correlators. For instance, let us consider the correlator

$$\langle TFD|\psi_L(u_1)\Pi_\kappa(0)^2\psi_L(u_1)|TFD\rangle.$$

This correlator corresponds to a situation when we have only two pairs of red dots on each side in the bottom of the Figure (6). Roughly speaking, after the measurement we immediately jump from + part of our Schwinger-Keldysh contour to the − part of the contour. It requires us to glue the Euclidean reparametrizations $f_+^\beta$ to $f_-^\beta$ at the time of a measurement $u=u^*$:

$$f_+^\beta(u^*)=f_-^\beta(u^*),\qquad f_+'^\beta(u^*)=f_-'^\beta(u^*),$$
$$f_+''^\beta(u^*)=f_-''^\beta(u^*)-2i\widetilde{\kappa}f_+'^\beta(u^*).\tag{48}$$

For the sake of brevity we will set $u^*=0$. The factor of two in $2\widetilde{\kappa}$ comes from the fact that there are two projections separating two Euclidean circles (note that in the denominator of (39) we have two consecutive insertions of $\Pi_\kappa(0)$). The choice of $A_+$ in (41) is a gauge choice, that determines when we transition to a Lorentzian signature. We can set $A_+=-ie^{-i\alpha\beta/4}$ to continue the trajectory into Lorentz spacetime at $u=0$ (that correspond to $f_+^\beta(0)=0$). For $f_-^\beta(u)$ we can adopt the ansatz similar to (41):

$$f_-^\beta(u)=-\frac{e^{-\alpha u}-iA_-e^{-i\alpha\beta/4}}{A_-e^{-\alpha u}+ie^{-i\alpha\beta/4}}.\tag{49}$$

Using the boundary conditions (48) we come to the following solutions for the reparametrizations

$$f_+^\beta(u)=i\frac{\sinh\left(\frac{\alpha u}{2}\right)}{\sinh\left(\frac{\alpha u}{2}+i\frac{\alpha\beta}{4}\right)},\qquad f_-^\beta(u)=-i\frac{\sinh\left(\frac{\alpha u}{2}\right)}{\sinh\left(\frac{\alpha u}{2}-i\frac{\alpha\beta}{4}\right)},\tag{50}$$

where the parameter $\alpha$ is determined from the equation

$$\alpha\cot(\alpha\beta/4)=\kappa\Delta\left(\frac{\alpha^2}{4\sin^2\left(\frac{\alpha\beta}{4}\right)}\right)^\Delta,\tag{51}$$

when $\kappa=0$, we can get that $\alpha=2\pi/\beta$. This solution is real on the thermal Euclidean circle (taking $f\to if$ and $u=i\tau$) but in Lorentzian signature it is complex.

Using the solutions $f_\pm^\beta(u)$ (50) we can evaluate the correlator:

$$\langle TFD|\psi_L(u_1)\Pi_\kappa(0)\Pi_\kappa(0)\psi_L(u_2)|TFD\rangle$$
$$=\left(\frac{\alpha^2/4}{-i\sinh(\alpha(u_1-u_2)/2)+\frac{2\widetilde{\kappa}}{\alpha}\sinh(\alpha u_1/2)\sinh(\alpha u_2/2)}\right)^\Delta.$$

This is exactly the Streicher–Qi answer at low temperatures [51].

Now let us consider the situation when after the measurement, we continue a bit into the Lorentzian time, that allows us to compute the correlator (38). Since the Lorentzian part lies after projections or measurements are performed, we must set $f_+(u) = f_-(u)$ to provide a proper cancellation along the Schwinger-Keldysh contour. The most general ansatz, that solved the equations of motions, would have the following form

$$f_+(u) = \frac{e^{-\gamma u} - 1}{Be^{-\gamma u} + C}, \tag{52}$$

where we have already used that $f_\pm^\beta(u^* = 0) = 0$. Now we need to set total $SL(2,R)$ charges to zero. Since we choose $t_L(u) = t_R(u)$, it implies immediately $Q_1 = Q_{-1} = 0$. The last constraint $Q_0 = 0$ leads to the following equation [40]:

$$Q_0[t(u)_L] = -\frac{t_L'''}{t_L'^2} + \frac{t_L''^2}{t_L'^3} - t_L' = 0, \quad t_L(u) = 2\arctan f_+(u). \tag{53}$$

This implies $F = \frac{1}{B}$. To determine the unknown parameters $B$ and $\gamma$ we use the boundary conditions:

$$f_+'(0) = f_+'^\beta(0), \quad f_+''(0) = f_+''^\beta(0) - i\widetilde{\kappa} f_+'^\beta(0). \tag{54}$$

since we encounter only one projection we should set just $\widetilde{\kappa}$ in the boundary conditions in comparison to the equation (48). Taking into account the equation (51) we arrive at the following solution in the Lorentzian time

$$f_+(u) = \tanh\left(\frac{\widetilde{\alpha}u}{2}\right), \quad \widetilde{\alpha} = \frac{\alpha}{\sin\left(\frac{\alpha\beta}{4}\right)}. \tag{55}$$

This solution is real and the $u$ dependence is controlled by $\alpha/\sin(\alpha\beta/4)$ instead of just $\alpha$, that effectively changing the temperature and energy of the system. Now we can compute the correlator (38):

$$\langle TFD|\psi_L(u_1)\Pi_\kappa(0)\psi_L(u_2)\Pi_\kappa(0)|TFD\rangle = \left(\frac{\alpha^2/4}{M}\right)^\Delta,$$
$$M = -i\sinh(\alpha u_1/2)\cosh(\widetilde{\alpha}u_2/2) + i\sinh(\widetilde{\alpha}u_2/2)\cosh(\alpha u_1/2)\sin(\alpha\beta/4)$$
$$+ \frac{\widetilde{\kappa}}{\alpha}\sin(\alpha\beta/4)\sinh(\widetilde{\alpha}u_2/2)\sinh(\alpha u_1/2), \tag{56}$$

Finally we are ready to study the anticommutators. First we need to analytically continue $u_2 \to -i\beta/2 - u_2$ and multiply the final expression by $i^{-\Delta}$ to convert $\psi_L(u_2)$ to $\psi_R(u_2)$. Then a quick examination reveals that the anti-commutator is maximal for $u_1 = -u_2 = u$. This is similar to the GJW teleportation. Figure 7 shows a few sample plots. We can estimate that at late times (larger than the thermal time $\beta$) and small $\kappa$, $G_{LR}$ behaves as

$$G_{LR} \approx \left(\frac{\alpha^2/4}{1 - i\frac{\widetilde{\kappa}}{\alpha}\sinh(\alpha u/2)\cosh(\alpha u/2)}\right)^\Delta. \tag{57}$$

GJW practitioners can recognize here the answer after an instantaneous insertion of $\mu\mathcal{O}_L\mathcal{O}_R$ in the Hamiltonian *except* that here we have $i\widetilde{\kappa}$ instead of $\mu$. Because of that the correlator never blows up. It means that the transition rate is suppressed by $\alpha^{2\Delta} \sim \beta^{-2\Delta}$. In the GJW case the naive blow-up behavior signals the breakdown of low-energy approximation (as $\psi_L, \psi_R$ can cross each other's lightcone) and the actual value of $G_{LR}$ is of order 1.

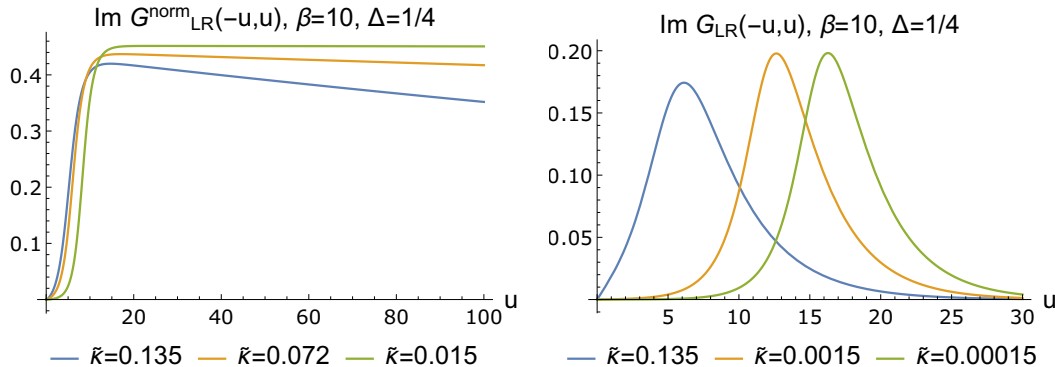

Figure 7: Left: Anti-commutator (39) with alternative normalisation. Right: anti-commutator (38).

Naively, the teleportation is possible for both signs of $\widetilde{\kappa}$. However, for negative $\widetilde{\kappa}$, correlator $\langle TFD|\psi_L(u_1)\Pi_\kappa(0)\Pi_\kappa(0)\psi_L(u_1)|TFD\rangle$ blows up for large $u_1$. It happens because operator $S$ in the weak projector $e^{-\kappa S}$ measures the operator size growth of $\psi_L(u_1)$. So one is restricted to $\widetilde{\kappa} > 0$.

Alternatively normalized correlator $G_{LR}^{\text{norm}}$ behaves as

$$G_{LR}^{\text{norm}} \approx \left( \frac{2\widetilde{\kappa}\sinh(\alpha u/2)^2/\alpha}{1 - i\frac{\widetilde{\kappa}}{\alpha}\sinh(\alpha u/2)\cosh(\widetilde{\alpha}u/2)} \right)^{\Delta}. \tag{58}$$

It is not suppressed by powers of temperature and can become of order 1. From Kitaev–Yoshida computation we expect that the information is transferable after SYK scrambling time. For small $\widetilde{\kappa} \ll 1$ (39) information gets transferred even earlier. Thus $G_{LR}^{\text{norm}}$ reaches maximal value at $u \sim -\beta \log \widetilde{\kappa}$. Eventually it decays to zero because $\widetilde{\alpha} > \alpha$. However, for small $\widetilde{\kappa}$ there is a long plateau where it reaches the maximal value of $2^{\Delta}\sin(\pi\Delta/2)$. It principle, $\widetilde{\kappa}$ can be made as small as $1/N$. Then the transfer starts after the scrambling time and continues for times of order $N$.

### 4.3 Multiple projections: Creating a wormhole

Now we want to make a step further and consider multiple consecutive measurements at different times. The setup is the same as in Figure 6, but we have many projections (red dots). As in the previous section we consider the situation when we prepare the system in the TFD state and then switch on continuous projections at time $u = 0$.

Recall that our weak projections act in the following way (18):

$$\widetilde{\rho} \to \dots e^{-\kappa S\Delta u}e^{iH\Delta u}\widetilde{\rho}e^{-\kappa S\Delta u}e^{-iH\Delta u}\dots \tag{59}$$

Basically it leads to an interaction similar to MQ interaction [40], *except* the signs on $\pm$ parts of the SK contour are different and purely imaginary. Nonetheless, this is consistent with having complex-conjugate solutions $f_+(u) = f_-^*(u)$.

Because of that, we can still use the formalism developed in Section 3, but we should modify it slightly. We reintroduce the variable $\phi(u)$ that is related to the reparametrization $f(u)$ in Poincare time as

$$e^{\phi} = t' = (2\arctan(f))' = \frac{2f'}{1+f^2}. \tag{60}$$

On Lorentzian parts of the Schwinger-Keldysh contour (the spikes on Figure 4) the reparametrization should satisfy the zero charge conditions and we arrive at the following equation (compare to the eq. (29)):

$$\phi''_\pm = -e^{2\phi_\pm} \pm i\kappa\Delta e^{2\Delta\phi_\pm}. \tag{61}$$

Note that in this equation the $\kappa$ appears instead of $\widetilde{\kappa}$ in comparison to the boundary conditions (47). Thermal solution on the Euclidean (semi-)circle is given by eq. (41). Also we want to satisfy the following boundary conditions:

- Continuity at $u = 0$, when we glue the real-time solution to the thermal one:

$$f_\pm(0) = f_\pm^\beta(0), \quad f'_\pm(0) = f'^\beta_\pm(0), \quad f''_\pm(0) = f''^\beta_\pm(0), \tag{62}$$

  Note that since we introduce continuous measurements the second derivative also should be continuous.

- Gluing together $f_+(u)$ and $f_-(u)$ at final time $T$, when we are done with the measurements:

$$f_+(T) = f_-(T), \quad f'_+(T) = f'_-(T), \quad f''_+(T) = f''_-(T), \tag{63}$$

  that could be thought as a requirement of $f_+(u) = f_-(u)$ in a consequent real-time evolution.

The boundary condition at $T$ then implies that $\phi_\pm(T)$ and $\phi'_\pm(T)$ are real. The above patching procedure can be done numerically. Since $\phi^*_+(u) = \phi_-(u)$, we can focus only on the plus field $\phi_+(u)$. First we fix $T$, the time when we $\pm$ branches merge. The thermal boundary conditions for $\phi$ at $u = 0$ are parameterized by only $\alpha$ ($A_+$ drops out once we convert $f$ to $\phi$). We need to find complex $\alpha$ such that $\phi_+(T), \phi'_+(T)$ are real. That will provide two real equations for two real parameters, that could be in principle solved by using shooting method or some more advanced tools. That will allow us to completely determine the evolution of the system. Equivalently, we can fix $\alpha$ and just solve the equations of motion, and see whether the trajectory terminates on a real line or not. That would provide us with two functions $T(\alpha), \phi_+(\alpha)$, that allows us to compute all interesting correlation functions.

For $\Delta = 1/2$ the solution can be obtained analytically:

$$\exp(\phi_\pm) = \frac{2v_\pm \exp(v_\pm(u - s_\pm))}{1 \mp 2\eta i \exp(v_\pm(u - s_\pm))/v_\pm - \kappa^2 \exp(2v_\pm(u - s_\pm))/v_\pm^2 + \exp(2v_\pm(u - s_\pm))}. \tag{64}$$

This solution depends on two parameters $v_\pm, s_\pm$, that are complex conjugated to each other $v^*_+ = v_-, s^*_+ = s_-$. With the use of boundary conditions at $u = 0$ we can find this parameters. By going numerically through different $\beta$ and $\kappa$ we find that increasing $\kappa$ does not lead to qualitative differences: left-right two-point function $G_{LR}$ always behaves like on Figure 2. That is, there is no phase transition: after turning on $\kappa$ at $u = 0$ one can transfer information, but the efficiency decays exponentially. In this trivial phase $\phi_\pm$ trajectories just run away to infinity. Also $G_{LR}(T, T)$ decays exponentially signalling the growth of the Einstein–Rosen bridge and losing correlation between left and right.

For generic $\Delta$ one has to resort to numerics. Interestingly, for $\Delta < 1/2$ and large enough $\kappa$ we do find a wormhole solution. For $\Delta = 1/4$ a sample (complex) trajectory $\phi_+(u)$ in this new phase is shown on Figure 8. We see that at late times the trajectories spend at lot of time around the critical point $\phi^*_+$ and then reach the same point $\phi_\infty$ on the real axis, where it should be glued to $\phi_-$. This results in constant $G_{LR}(T, T) \sim e^{2\Delta\phi_\infty}$ which says that the Einstein–Rosen bridge has stopped growing - the right panel of Figure 8. Also now one can teleport information

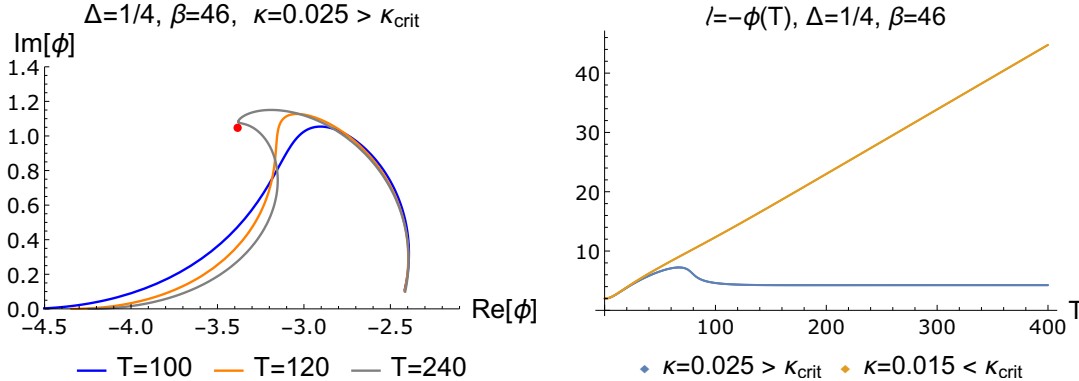

Figure 8: Left: Trajectory $\phi_+(u)$ in the wormhole phase. Red dot is the equilibrium saddle point. For longer evolution times $T$, the trajectory spends more and more time around the saddle point. Right: wormhole length $\log G_{LR}(T, T)$ as a function of time in different phases. The fact that it is constant in the wormhole phase reflects the fact that $\phi_+$ ends up in the same point on the real axis on the left plot. For $\beta = 46$ the critical coupling $\kappa_{crit} \approx 0.02$.

at any time - the left panel of Figure 3. Anti-commutator $\operatorname{Im} G_{LR}(u_1, T)$ eventually decays to zero for large $T$, but it does not really matter when (at what $u_1$) you inserted the perturbation. The decay to zero can also be easily explained by the fact that trajectories spend most of the time around the saddle point. Consider the equation (28) for $G_{LR}$:

$$G_{LR} = \frac{t_L'(u_1)^\Delta t_L'(u_2)^\Delta}{\cos^{2\Delta}\left(\frac{t_L(u_2) - t_L(u_1)}{2}\right)}. \tag{65}$$

For a fixed $u_1$ and large $u_2$, the difference $t_L(u_2) - t_L(u_1) = \int_{u_1}^{u_2} du\, e^\phi$ is dominated by the saddle hence the integral is equal to $e^{\phi_*}(u_2 - u_1)$. The value of $\phi_*$ can be easily determined from eq. (61):

$$e^{2\phi_*} + i\kappa\Delta e^{2\Delta\phi_*} = 0. \tag{66}$$

The value of $t_L'(u_2)$ is determined by the position of the trajectory $\phi(u)$ where the $+$ Schwinger-Keldysh part meets the $-$ part and $\phi$ is real. At late times this position is a constant, since the trajectory has almost zero velocity near the critical point, so it intersects the real axis almost at the same point - Figure 8. Hence, the Green function can be approximated as

$$G_{LR} = \text{const} \frac{1}{\cos^{2\Delta}\left(e^{\phi_*}\frac{u_2 - u_1}{2}\right)}, \tag{67}$$

this is the result quoted in the Introduction. It does exhibit revivals, but since $e^{\phi_*}$ is complex, $G_{LR}$ eventually decays to zero. In the GJW/MQ case, the Green functions has exactly the same form but $e^{\phi_*}$ is real hence the behavior is purely oscillatory. Figure 9 zooms in on a late time behavior of the $G_{LR}$ in the KY case.

Using the fact that the trajectory at late times spend most of the time around the critical point also explains why all trajectories terminate at the same point $\phi_\infty$. The equations of motions for the fields $\phi_\pm$ (61) has a conserved metric

$$E_\pm = \phi_\pm'^2 + e^{2\phi_\pm} \mp i\kappa e^{2\Delta\phi_\pm}, \tag{68}$$

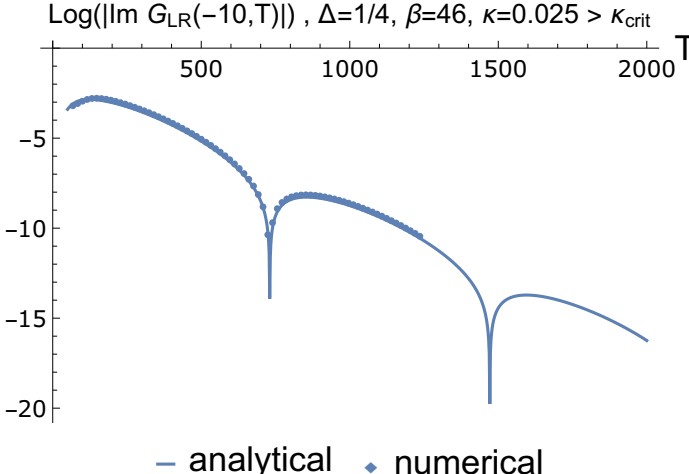

Figure 9: Late time behavior of $G_{LR}$. Since it is exponentially decaying it is convenient to plot $\log|\operatorname{Im}G_{LR}|$. The analytical answer is given by (67).

we can see that as $T$ tends to infinity the trajectories stays around the critical point $\phi_{\pm}^{*}$ closest to the real axis

$$-e^{2\phi_{\pm}^{*}} \pm i\kappa\Delta e^{2\Delta\phi_{\pm}^{*}} = 0, \quad \phi_{\pm}^{*} = -\frac{\log(\pm i\kappa\Delta)}{2(\Delta-1)}, \tag{69}$$

because only around this point the trajectory can stay infinitely long. The provides us with some approximate energy $E_{\pm}^{*}$. After that it easy to see that if the trajectory terminated at the real axis at $\phi_{\infty}$ the following equation must hold

$$\operatorname{Im}E_{+}^{*} = \kappa e^{2\Delta\phi_{\infty}}, \quad \phi_{\infty} = \frac{1}{2\Delta}\log\left[\frac{1}{\kappa}\operatorname{Im}E_{+}^{*}\right]. \tag{70}$$

Finally, lets us comment on the steady-state solution. Our numerical analysis suggests that in the wormhole phase the system does not depend on the initial temperature - Figure 10. It means that the maximal value of the teleportaion fidelity $\operatorname{Im}G_{LR}$ depend only on $\kappa$.

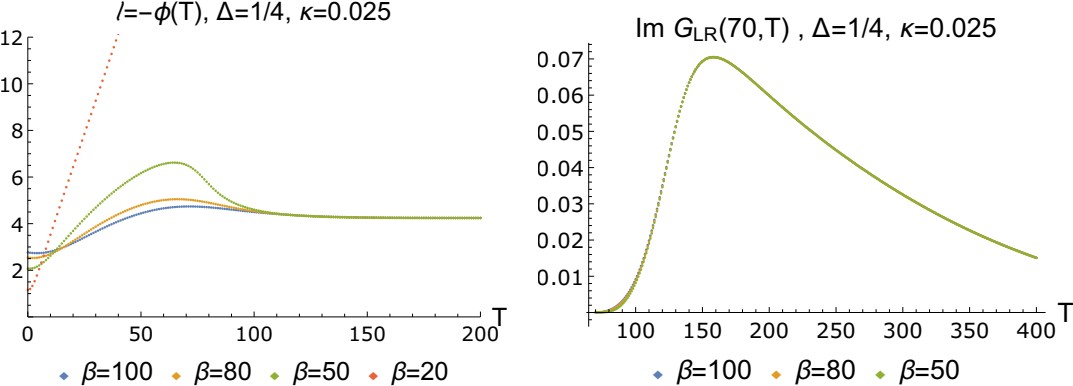

Figure 10: The steady state in the wormhole phase does not depend on the initial temperature $\beta$ of TFD. Left: behavior of $\phi(T)$ for different initial temperatures, Right: two-sided correlation function.

# 5   Conclusions and discussion

This paper has two major conclusions. First, projections/measurements could be approximated with a low-energy quantum channel. Then one can perform multiple measurements without introducing too much energy into the system and hence study phase transitions in observables linear in the density matrix. Also this opens a possibility of studying unitary-measurement dynamics in continuum QFT.

Second, in SYK/JT gravity there is a phase transition in teleportation rate triggered by continuous projection dynamics. In geometric terms it is a transition between TFD state (two entangled black holes) and an eternal traversable wormhole. In the KY case it is a novel type of wormhole supported by projections rather than negative energy.

This techniques could be generalized to higher dimensions. The interaction (19) could be generalized to any continuous QFT.It would be interesting to see if it can support a wormhole in higher dimensions. Also in Section 4.3 we presented specific examples of wormhole solutions supported by projections, but we do not have as much analytic control as in the GJW case. We have only numerical evidence that the transition happens for $\Delta < 1/2$. It would be interesting to understand why it happens and determine $\kappa_{crit}$. The biggest challenge is the inherent time-dependence of the solution: it is true that the trajectory $\phi_+$ spends a lot of time around the saddle point but eventually it must come to real axis to meet with $\phi_-$. Also since the dynamics is happening around a saddle rather than a minimum, this raises concerns about its stability. We leave this question for future work.

Another important question is the nature of teleportation. In case of the GJW protocol with a single unitary insertion, one one easy explain teleportation using negative energy pulses in the dual gravity picture [36]. From the boundary quantum mechanics perspective the GJW protocol works due to the specific distribution of phases (perfect size-winding) in the operator spreading dynamics. Perfect size-winding is expected for holographic theories, but not for general chaotic Hamiltonians. It would be interesting to generalize the arguments of [34] to the continuous GJW setting and explain the phase transition. The KY case is different. The KY protocol with a single projection is guaranteed to work for any chaotic Hamiltonian. Dual gravity picture in this case is not clear. Comparing (68) and (30) we can naively conclude that KY inserts "complex" energy into the bulk. In this paper we justified using complex reparametrizations (and hence complex bulk geometry) by reproducing in Section 4.2 a known large-q SYK answer this way. Moreover, in case of single projection one can perform all computations in Euclidean where everything is real. Complex geometries in gravity are not something exotic and recently they received a new wave of attention [52–56]. Our situation is reminiscent of replica wormholes: in Euclidean the geometry is real [57], but in Lorentz signature it becomes complex [58,59]. It would be interesting to explore the complex geometry of the KY teleportation further.

The biggest issue with projections is a post-selection procedure: one has to do a measurement first and then discard the outcomes in which system ends up not in the state we want. It leads to a problem, that the number of states we can work with is exponentially small. A natural thing to do then is not to discard that sample, but to try to push the system towards what we want [7,60–64]. We can try to follow this approach in our case. Returning to eq. (5), if the measurement of $i\psi_L^j \psi_R^j$ yields $+1$, we act with another $\psi_L$ to turn it into $-1$ eigenstate:

$$\rho \to (1 - 4N\kappa du)\rho + \kappa du \sum_j (1 - i\psi_L^j \psi_R^j)\rho(1 - i\psi_L^j \psi_R^j)$$

$$+ \kappa du \sum_j \psi_L^j (1 + i\psi_L^j \psi_R^j)\rho(1 + i\psi_L^j \psi_R^j)\psi_L^j. \tag{71}$$

Now the trace is preserved. Quartic terms in the action are problematic. Inspired by Section

2, we can try to drop them and consider the action

$$\frac{1}{\kappa}\partial_u\rho = -i\psi_L^j\psi_R^j\rho - \rho i\psi_L^j\psi_R^j + i\psi_R^j\rho\psi_L^j - i\psi_L^j\rho\psi_R^j. \tag{72}$$

It is trace-preserving, but unlike eq. (19) it is not completely positive (not even positive). It would be interesting to come up with another notion of a driving which does not have this problem. In principle, dynamics (71) can be studied numerically, but one would not be able to do it within the low-energy Schwarzian approximation.

## Acknowledgments

We would like to thank M. Tomasevic for collaboration at early stages of this project. Also we are grateful to E. Colafranceschi for reading the mansuscript and providing comments. It is pleasure to thank A. Gorsky, B. Grado-White, A. Khindanov, J. Maldacena, D. Marolf, H. Marrochio, V. Su, W. Weng, M. Usatyuk, Y. Zhao and especially S. Antonini, P. Glorioso and S. Diehl for discussions and comments. AM also would like to thank C. King for moral support.

**Funding information** F.K.P. is currently a Simons Junior Fellow at NYU and supported by a grant 855325FP from the Simons Foundation. This material is based upon work supported by the Air Force Office of Scientific Research under award number FA9550-19-1-0360. It was also supported in part by funds from the University of California. AM would like to acknowledge support from Berkeley Center for Theoretical Physics and by the Department of Energy, Office of Science, Office of High Energy Physics under QuantISED Award DE-SC0019380 while visiting Berkeley Center for Theoretical Physics.

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
