# Peer review of "Measurement-induced phase transition in teleportation and wormholes"

_SciPost Physics, doi:SciPost Phys. 17, 020 (2024)_

## Round 1 · Referee Report · Anonymous (Referee 1) · 2024-5-14

Report

Looks good! All of the comments in my initial report have been addressed.

Recommendation

Publish (easily meets expectations and criteria for this Journal; among top 50%)

---

## Round 1 · Referee Report · Anonymous (Referee 2) · 2024-5-27

Report

The authors have addressed all of the questions, and I now recommend this work for publication.

Recommendation

Publish (easily meets expectations and criteria for this Journal; among top 50%)

---

## Round 1 · Author Response

First of all, we thank the reviewers for their encouraging and detailed comments. We sincerely apologize for a long delay in our response. New text in the draft in marked in blue (for the editorial team: \new{} command in latex) Also we fixed a number of typos/grammatical mistakes (not marked).

Let us now address the specific questions. Most of these replies are reflected in the manuscript.

Both reviewer asked about the absence of revivals for the KY case:

We believe KY has revivals as well. Because of the decoherence coming from measurements they are damped with time, unlike the MQ case. Their presence can be inferred from the analytical approximation (4.30) in the new version of the paper. Basically, the KY wormhole solution is similar to the standard MQ wormhole solution except that the length $e^{\phi_*}$ is complex. Hence the two-point function

$$ 1/cos( e^{\phi_*} (t_2 - t_1))^{2\Delta} $$
not just oscillates, but is damped as well. In the previous version we did not study the numerical evolution for large enough times to see the revivals.

Reviewer 1:

“exponentially many measurements.” Exponentially many in what? This sentence is not clear to me. The better way to phrase this is exponentially many samples. The evolution with post-selection means that after performing a measurement, a sample with undesired measurement outcome is completely discarded. So, if the desired outcome has probability $p$, then simulating $t$ time-steps will roughly require $1/p^t$ samples.

On page 4, it would be good to clarify what you mean by “it requires a finite \mu coupling.” Does this mean the effect can’t be seen in > perturbation theory in \mu? The statement we are making is that for a fixed initial state the wormhole becomes eternally traversable only if \mu is large enough. In the paper we find the corresponding critical value analytically as a function of temperature - eq. (3.13) and below. Equations are simple enough to have an exact solution for any \mu. We believe the transition to an eternal wormhole cannot be seen in the perturbation theory in \mu because above the critical \mu the classical solution completely changes its behavior. The equations can be mapped to a classical particle moving in a potential. If \mu is large enough the particle trajectory becomes trapped rather than running away to infinity. Such qualitative change of behavior cannot be treated as a small perturbation.

Around equations 2.7-2.10 it isn’t clear what the authors mean by the “plus and minus parts,” and when they should be expected to “cancel out.” We added further explanations there. "Plus and minus" parts are literary forward $e^{-i H u}$ and backward $e^{i H u}$ time evolution operators in the Heisenberg-evolved density matrix $e^{-i H u} \rho e^{+ i H u}$.

Reviewer 2:

Section 3.1 contains the review of the Maldacena--Qi solution. In principle, their equations are valid even for time-dependent \mu, but they never studied them in this regime. Our contribution is Section 3.2, where we solve them after the \mu coupling is suddenly turned on. We have also added the clarification regarding finite \mu coupling.

Dynamics with post-selection is somewhat acausal: the system is conditioned to be in a certain state at t=0. Hence, the corresponding saddle-point in the path integral is modified both after and \textit{before} the projection. After a single projection the solution is the standard thermal solution - eq. (4.18). The projection, unsurprisingly, raises the temperature (since \tilde{\alpha} > \alpha). Before the projection the solution is given by (4.13). In the leading order correction coming from the projection strength $\kappa$ is purely complex, this is why it is hard to interpret it.

Is it straightforward to compare this to the MQ time-reversed problem The reverse MQ problem was studied in the original MQ paper. There nothing interesting happens: turning off \mu always leads to the standard TFD solution.

---

## Editorial Decision

published